# Biological Activity of Artificial Plant Peptides Corresponding to the Translational Products of Small ORFs in Primary miRNAs and Other Long “Non-Coding” RNAs

**DOI:** 10.3390/plants13081137

**Published:** 2024-04-18

**Authors:** T. N. Erokhina, D. Y. Ryazantsev, S. K. Zavriev, S. Y. Morozov

**Affiliations:** 1Shemyakin-Ovchinnikov Institute of Bioorganic Chemistry, Russian Academy of Sciences, 117997 Moscow, Russiaszavriev@ibch.ru (S.K.Z.); 2Biological Faculty, Belozersky Institute of Physico-Chemical Biology, Lomonosov Moscow State University, 119991 Moscow, Russia

**Keywords:** micropeptide, plants, transcription, lncRNA, miPEP, primary microRNA, functions of miPEPs, biotechnology

## Abstract

Generally, lncPEPs (peptides encoded by long non-coding RNAs) have been identified in many plant species of several families and in some animal species. Importantly, molecular mechanisms of the miPEPs (peptides encoded by primary microRNAs, pri-miRNAs) are often poorly understood in different flowering plants. Requirement for the additional studies in these directions is highlighted by alternative findings concerning positive regulation of pri-miRNA/miRNA expression by synthetic miPEPs in plants. Further extensive studies are also needed to understand the full set of their roles in eukaryotic organisms. This review mainly aims to consider the available data on the regulatory functions of the synthetic miPEPs. Studies of chemically synthesized miPEPs and analyzing the fine molecular mechanisms of their functional activities are reviewed. Brief description of the studies to identify lncORFs (open reading frames of long non-coding RNAs) and the encoded protein products is also provided.

## 1. Introduction

Long non-coding RNAs (lncRNAs), or long non-protein-coding RNAs (lnpcRNAs), having sizes of more than 200 nucleotides are commonly involved in many biological activities in eukaryotes and are previously thought to have no protein coding ability [1,2,3,4,5,6,7,8]. These RNAs are mostly transcribed in the nucleus with RNA polymerase II in an mRNA-like way. The mechanisms by which plant lncRNAs function are numerous, and some of them are better understood in animals [9,10,11,12,13,14,15,16]. Recent studies have shown that the short open reading frames (sORFs) of some plant lncRNAs encode functional small peptides (micropeptides) with a size of around 5 to 100 amino acids [10,12,17,18,19,20,21,22]. Importantly, these lncRNAs also include precursors of many microRNAs, which code for peptides called miPEPs (peptides encoded by primary microRNAs), and some trans-acting siRNAs [19,23,24,25,26,27,28,29,30,31,32,33,34]. Currently, bioinformatic tools as well as experimental methods such as mass spectrometry, ribosome mapping and *in vitro* translation are used to reveal the potential translational products of lncRNAs [10,16,17,19,21,24,32,35,36,37,38,39]. 

Generally, biological activity of the lncRNA-encoded peptides can be revealed, on the one hand, by genetic approaches. Using this strategy, investigation of possible peptide functions includes constructing mutants of a target small ORF. Genome modifications that disrupt an initiation codon or a coding sequence of such lncRNA ORF can be used to produce loss-of-function mutants of lncRNA-encoded peptide. Particularly, the CRISPR-cas9-mediated gene editing approach can be applied to check the potential activities of peptides. Alternatively, *in planta* overexpression of small peptides or expression driven by a native promoter, there may also be useful ways to study the lncRNA-encoded peptide functions *via* genetic approaches [17,24,40,41,42]. On the other hand, the use of artificially produced lncRNA-encoded peptides with a high purity and/or desired labeling provides an effective tool for studying plant small peptide functions. In principle, these peptides can be obtained by chemical synthesis. In addition, the *in vitro* translation and overexpression in microbe cells can be used (for example, see [17,24,42,43]). 

In this review, we mainly focus on the biological functions of artificial plant peptides corresponding to those encoded by regulatory ncRNAs, including precursors of microRNA (miPEPs). Here, we discuss the available data on the interaction of lncRNA-encoded peptides with nucleic acids, proteins, subcellular domains and plant tissues to help our understanding of peptide-based molecular networks within plant cells. Furthermore, we summarize the physiological effects and modes of action of the exogenously applied peptides in plant cells. Additionally, potential applications of chemically synthesized lncRNA-encoded peptides (mostly, miPEPs) in the field of plant biotechnology will be discussed.

## 2. Possible Effects of lncRNA-Encoded Peptides on Plant Physiology and Development

### 2.1. “Conventional” lncRNAs

ENOD40 gene is expressed as a transcript representing one of the first lncRNAs discovered in plants. It has been initially found that this lncRNA in soybean regulates the formation of symbiotic nitrogen-fixing nodules in association with soil rhizobial bacteria. It is also involved in the stimulation of colonization of plant roots by fungi [44]. In addition to plants of the order *Fagales*, homologs of this RNA have been identified in monocots such as rice and maize, indicating conservation of its molecular targets between dicots and monocots [35,45]. In contrast to legumes, rice OsENOD40 is specifically expressed in rice stems and is involved in organ differentiation and vascular tissue development [46]. In soybeans, ENOD40 directs the synthesis of 12- and 24-amino-acid long peptides that are encoded by short ORF1 and ORF2, respectively. Alfalfa ENOD40 ORF1 and ORF2 genes encode short peptides of 13 and 27 amino acids [35,45]. ENOD40 region coding for a short peptide of 12–13 amino acids is also conserved in many, but not all, non-leguminous species [45]. Translation *in vitro* of soybean ENOD40 RNA has been demonstrated to produce both ORF1 and ORF2 peptides, and these chemically synthesized peptide products have been found to bind to sucrose synthase [35]. It seems that the mechanisms of ENOD40 gene related to nodule formation may act *via* its RNA transcript molecule, and its encoded peptides may be responsible for some unknown functions only required in some specific legume plants [45].

The POLARIS gene (*PLS*) has been found in Arabidopsis to direct the synthesis of an auxin-inducible short “non-coding” transcript (~500 nucleotides in length). Plant seedlings homozygous for the T-DNA insertion in *PLS* have been shown to have reduced sizes compared with that in the wild type, and the length of the mutant primary roots is ∼50% of that of the wild type [47]. Sequencing of the *PLS* locus showed that the T-DNA inserted into the 25th codon of a short ORF encoding a predicted 36 amino acid residue peptide with no significant homology with known proteins [47] disrupts correct root growth and vascular development. Mutational analysis also implicates a role for the encoded peptide in ethylene signaling, auxin homeostasis and microtubule cytoskeleton dynamics [48]. However, no data with exogenic chemically synthesized PLS peptide has been published. 

Rotundifolia4-1D gene (*ROT4*) has been isolated during tagging screening of T-DNA mutants with phenotypes showing short leaves and a reduced number of cells in the proximodistal axis [49]. In these T-DNA insertion mutants, ROT4 ORF encoding a small protein of 53 amino acid residues is overexpressed. ROT4 is a member of a novel gene family, which shares a highly conserved internal 29 amino acid sequence. The constructs directing modified proteins, that lack either the N-terminal or the C-terminal regions of ROT4 peptide but retain highly conserved internal regions, have been expressed under the control of the CaMV 35S promoter in transgenic Arabidopsis plants. In both cases, the resulting transgenic plants had short rosette leaves similar to those of rot4-1D mutants. These data suggest that only the conserved peptide domain is required for ROT4 activity [49]. ROT4-GFP fusion protein is present in the plasma membrane and not in the cell wall. Summarized results suggest that ROT4 could be involved in the determination of organ boundary and positional cues of leaf primordia along the proximodistal axis [49,50].

T-DNA insertion mutagenesis in Arabidopsis has also been used for the identification of small ORFs in two provisionally non-coding RNAs encoding regulators of developmental cell death such as INFLORESCENCE DEFICIENT IN ABSCISSION (IDA) [51] and KISS OF DEATH (KOD) [52]. *IDA* gene encodes a small protein of 77 amino acids with an N-terminal signal peptide. This protein is suggested to participate in an ethylene-independent developmental pathway that controls floral abscission [51]. *KOD* gene codes for a peptide of 25 amino acids, and its expression is sufficient to cause cell death in leaves or seedlings and to activate caspase-like activities [52]. 

Multiple lncRNA-encoded peptides have been found not only in flowering plants but also in lower-land plants. During studies of moss *Physcomitrella patens*, nine translated short ORFs located on lncRNAs have been revealed [40]. Three of these ORFs, Pp3c18_sORF57 coding for 40 aa peptides, Pp3c9_sORF1544 (41 aa) and Pp3c25_sORF1000 (61 aa) are common between lncRNAs found in all moss cell types, and the translation of the corresponding peptides is confirmed by mass spectrometry. Overexpression and knockdown studies of these lncRNA-encoded peptides have revealed morphological variations showing their role in moss growth and development: (i) knocking out a 40-aa peptide encoded by lncRNA Pp3c18_sORF57 showed a slight decrease in moss plant diameter on medium with glucose and without ammonium tartrate. Moreover, no changes have been observed in protonemal architecture and the number of leafy shoots or filament branching in knockouts; (ii) overexpression of a 41-aa peptide encoded by the Pp3c9_sORF1544 results in longer filaments implicated in a rapid radial extension of the protonemal tissues compared to the wild-type lines; (iii) knocking out a 61-aa peptide encoded by conserved Pp3c25_sORF1000 results in a decrease in growth rate and altered protonemal architecture on medium without glucose but supplemented with ammonium tartrate; (iv) the lines with a knockout in a 57-aa peptide encoded by an additional conservative lncRNA Pp3c25_sORF1253 display a decrease in protonemata growth rate and altered filament branching [40,42]. 

### 2.2. Chemically Synthesized Plant miPEPs 

#### 2.2.1. Family *Fabaceae*

One of the first miPEPs identified in plants is miPEP171b from *Medicago truncatula,* which is encoded by pri-miR171b (primary microRNA171b). This precursor transcript of miRNA contains an ORF of 20 codons located upstream of the hairpin pre-miRNA (precursor microRNA) region [53]. In plants, overexpression of Mt-miPEP171b reduces lateral root density. Likewise, the addition of exogenous Mt-miPEP171b to *M. truncatula* seedlings results in a reduction of lateral root formation, and like *in planta*, overexpressing enhances the abundance of miR171b (seemingly synthesis of pri-miRNA) [53]. Interestingly, exogenous Mt-miPEP171b increases the expression of other plant genes in addition to its own pri-miRNA. This gene represents *LOM1* (LOST MERISTEMS 1) encoding transcription factor belonging to the GRAS family [54]. Moreover, Mt-miPEP171b stimulates arbuscular mycorrhization [54]. Importantly, short ORF located in *M. truncatula* pri-miR171b downstream of Mt-miPEP171b ORF has 5 codons in length, and the corresponding small synthetic peptide Mt-miPEP171b2 activates transcription of pri-miR171b in heterologous plant system like longer miPEP171b (*Nicotiana benthamiana*) [55] (Table 1). 

It is quite interesting that other exogenous Mt-miPEP171 “a” (10 amino acids in length), “c” (7 residues), “d” (6 residues), “e” (23 residues) and “f” (5 residues), which are sequence unrelated to Mt-miPEP171b, decrease mycorrhization rate and LOM1 expression [54]. 

Like Mt-miPEP171b of *M. truncatula*, Lj-miPEP171b (22 amino acids) from *Lotus japonicus* increases expression of pri-miR171b and mycorrhization when it is exogenously applied to *L. japonicus* plants [54]. 

Soybean (*Glycine max*) can fix atmospheric nitrogen through their symbiotic interaction with soil bacteria commonly known as rhizobia. This interaction takes place in symbiosis-specific organs, the nodules. It has been shown that exogenous Gm-miPEP172c, having 16 amino acids in length, stimulates pri-miR172c expression and significantly increases the nodule number observed in plants without affecting root development [56]. 

Recently, it has been demonstrated that the exogenous application of some *Phaseolus vulgaris* miPEPs to bean leaves results in significant resistance to necrotrophic fungus *Botrytis cinerea*. These sequence-unrelated peptides include Pv-miPEP169h (23 amino acids in length), Pv-miPEP169k (11 amino acids) and Pv-miPEP169p (24 amino acids) [57] (Table 1). 

#### 2.2.2. Family *Sapindaceae*

Studies of miR166 gene functions during the early somatic embryogenesis in *Dimocarpus longan* [58] have shown that longan Dl-miPEP166 S338 (50 amino acids in length) increases the expression of pri-miRNA and inhibits plants gene *ATHB15* coding for transcription factor (Table 1). 

#### 2.2.3. Family *Brassicaceae*

It has been indicated in an earlier paper [53] that pri-miRNAs of many *Arabidopsis thaliana* microRNAs contain short ORFs in the 5′-proximal regions. These ORFs encode miPEPs of 3 to 59 amino acids long and show no significant sequence similarity (Table 1). This fact suggests that each miPEP is specific for its corresponding miRNA. One of the predicted *A. thaliana* miPEPs (At-miPEP165a) is well conserved among the plants of the family *Brassicaceae* and has 18 amino acids in length [53]. The expression of pri-miR165a has been measured in *A. thaliana* seedlings treated with exogenous miPEP165a. This treatment increases the amount of pri-miR165a, whereas the presence of RNA-polymerase inhibitor cordycepin completely abolishes the positive effect of miPEP165a. These facts suggest that miPEP activates the transcription of pri-miRNAs [53]. Additional experiments have demonstrated that the increase in primary root length induced by the chemically synthesized At-miPEP165a treatment is due to the stimulation of cellular proliferation rather than an increase in cell length [53,59]. However, most chemically synthesized At-miPEPs have an inhibitory effect on primary root growth. These peptides include: At-miPEP157b; At-miPEP157c; At-miPEP159b; At-miPEP159c; At-miPEP163; At-miPEP164b; At-miPEP164c; At-miPEP166a; At-miPEP169a; At-miPEP171c; At-miPEP390a; At-miPEP391; At-miPEP393b; At-miPEP394b; At-miPEP395a; At-miPEP395c; At-miPEP395f; At-miPEP396a; At-miPEP398c and At-miPEP399d. Peptides with stimulation effects on total root development are represented by At-miPEP166g and At-miPEP397a [60]. Importantly, both primary-root-stimulating At-miPEP397a and inhibitory At-miPEP164b can increase the expression of their cognate pri-miRNAs [60]. Recently, additional chemically synthesized miPEPs (At-miPEP156a, At-miPEP162a, At-miPEP162b, At-miPEP163, At-miPEP167a, At-miPEP169l, At-miPEP172b and At-miPEP396a) capable of stimulating synthesis of own pri-miRNA have been revealed [55]. 

One more miPEP of *A. thaliana* studied in some detail is At-miPEP858a. The corresponding pri-miR858a contains three putative ORFs in a region upstream from the pre-miR858a. The fusion expression of the GUS reporter gene with the translation initiation codon shows that only ORF1 (44 codons) is translationally active and gives rise to At-miPEP858a [41]. It has been revealed that chemically synthesized miPEP858a, like many other *A. thaliana* miPEPs, modulates the accumulation of mature miRNA by enhancing the transcription of the corresponding pri-miRNA. Accordingly, At-miPEP858a increases the expression of miR858a; thus, this decreases the expression of the target genes, leading to the modulation of associated phenotypes, namely, changing plant development (increase of root length) and modulated levels of flavonoids [41]. Moreover, the application of synthetic At-miPEP858a to growing *A. thaliana* plants results in early bolting and a significant increase in plant height compared to plants treated with water. It has been proposed that general miPEP858-dependent modulation of plant development might be regulated by the inhibition of auxin transport due to changed levels of flavonoids [41]. At-miPEP171c also influences plant development. The growth of primary roots of young plants treated with At-miPEP171c is inhibited significantly. In contrast, the number of lateral roots and adventitious roots increases significantly in the early stage of root development [63].

It has been found recently that the family *Brassicaceae* miPEP from one species can activate the transcription of pri-miRNAs from other species [55]. Particularly, chemically synthesized At-miPEP156a (33 amino acids in length), being the highly conservative among miPEPs of *Brassicaceae* [66], can significantly stimulate transcription of pri-miR156a in *Brassica oleraceae* and *Brassica rapa* [55] (Table 1). We have previously revealed that two last species also encode miPEPs (Bo-miPEP156a and Br-miPEP156a), through which exogenous application to Brassica seedlings increases the main root length [61]. Importantly, chemically synthesized Bo-miPEP156a stimulates the transcription of pri-miR156a in *A. thaliana* [55]. 

Among *Brassicaceae*, two additional poorly conserved miPEPs (Bo-miPEP397a and Bv-miPEP164b) have been revealed in *B. oleraceae* and *Barbarea vulgaris* [60]. Both miPEPs can increase the expression of their respective pri-miRNAs. However, exogenous application (watering) of Bo-miPEP397a (10 residues) increases total plant size (a 36% increase in the foliar surface), whereas Bv-miPEP164b (8 amino acids) results in a 21% decrease in the foliar surface [60]. 

Very recently, three *A. thaliana* miPEPs have been shown to induce significant resistance to necrotrophic fungus *Botrytis cinerea*. These miPEPs include At-miPEP169c, At-miPEP169h and At-miPEP396b, and all of them stimulate the synthesis of the respective miRNAs [57]. 

*A. thaliana* miPEP regulating abiotic stress response has also been described [62]. At-miPEP408 (35 aa in length) is capable of a concentration-dependent increase in the accumulation of At-pre-miR408. The addition of the synthetic peptide to the growth media increases the sensitivity of seedlings toward low sulfur and arsenite As(III) stresses. This is evident from the greater reduction in primary root length and fresh weight. The additional results suggest that the peptide induces a higher accumulation of ROS and, as a result, higher sensitivity to the abiotic stress [62].

#### 2.2.4. Family *Vitaceae*

Grapevine (*Vitis vinifera*) encodes several characterized miPEPs (Table 1). The first reported miPEP is Vvi-miPEP171d1, which includes seven amino acids [63]. Treatment of grape tissue culture plantlets with synthetic Vvi-miPEP171d1 results in the increased expression of Vvi-miR171d and pri-miR171d, and this effect is higher with treatment duration. However, the expression levels of other Vvi-miRNAs are unaffected [63]. The addition of synthetic Vvi-miPEP171d1 to cultured grapevine plantlets placed on IAA hormone-containing medium results in an increasing number of adventitious roots and a concomitant decrease in root length. Interestingly, culturing Arabidopsis plants with the application of synthetic Vvi-miPEP171d1 shows little effect on plant development. These results indicate that Vvi-miPEP171d1 has no activity in Arabidopsis, which suggests the functional specificity of chemically synthesized miPEPs upon application to distant plant species [63]. 

Multistep bioinformatic analysis has led to the identification of an additional small ORF in grapevine pri-miRNAs encoding a putative miPEP of 16 amino acids called Vvi-miPEP164c [64]. In grapevine cell suspensions 10 days after treatment with this chemically synthesized peptide, the transcript levels of pre-miR164c have upregulated, reaching a maximum effect at peptide concentration 1 μM (3.5-fold increase). Accordingly, enhancing the accumulation of miR164c leads to pronounced post-transcriptional gene silencing of transcription factor VviMYBPA1 and MYBPA1-activated genes. Such molecular events result in **the** inhibition of proanthocyanidin synthesis and the simultaneous increase of anthocyanin synthesis [64].

Recently, two more grape miPEPs, having activities in modulating plant stress response, have been determined [65]. To explore miRNAs potentially contributing to low-temperature tolerance of *V. vinifera* plantlets, the expression pattern of different Vvi-miRNAs has been analyzed under cold stress. Pri-miRNAs of two miRNA genes Vvi-MIR172b and Vvi-MIR3635b under stress conditions encode miPEPs, which can increase transient expression of corresponding miRNAs. These peptides are named Vvi-miPEP172b (16 amino acids) and Vvi-miPEP3635b (11 amino acids), respectively. If the synthetic Vvi-miPEP172b and Vvi-miPEP3635b have been applied to the grape plantlets, the plant tissue exhibits a higher cold tolerance compared with the controls [65]. 

#### 2.2.5. Family *Solanaceae*

Interesting data on the miPEPs influencing biotic stress response have been recently published [57]. In tomato (*Solanum lycopersicum*), lesion size on plants inoculated with *B. cinerea* spores and treated with synthetic peptide Sl-miPEP169d (10 amino acids in length) is smaller than in the control, whereas peptides Sl-miPEP169j (22 amino acids) and Sl-miPEP396d (37 amino acids) cause increasing infection-induced lesion size. Moreover, exogenic application of Sl-miPEP169d lowers the level of tomato leaf infection by *Alternaria solani*, a fungus responsible for early blight, and gives rise to the protective effect on the tomato against bacteria *Pseudomonas syringae* and *Xanthomonas* sp. in field trials [57]. This effect concerns the total number of fruits obtained, and in total, SlmiPEP169d can improve tomato crop yields by 35%.

#### 2.2.6. Family *Rosaceae*

The experiments with the synthetic miPEPs influencing biotic stress response and found in strawberries (*Fragaria vesca*) have also identified peptides capable of inducing biotic stress resistance [57]. Four peptides, namely, Fv-miPEP169h (21 aa), Fv-miPEP169l (23 aa), Fv-miPEP396a (15 aa) and Fv-miPEP396f (18 aa) can decrease infection-induced lesion size, whereas two peptides, Fv-miPEP169g (13 aa) and Fv-miPEP169j (21 aa), significantly increase lesion size after leaf inoculation with *B. cinerea* spores [57]. 

#### 2.2.7. Family *Poaceae*

Recently, a novel regulatory peptide (Os-miPEP156e), which is functional in abiotic stress response, has been discovered in rice (*Oryza sativa*) [67] (Table 2). This peptide (36 amino acids in length), when overexpressed in rice, is capable of enhancing the transcription level of pri-miR156e and miR156e. Moreover, exogenous synthetic Os-miPEP156e significantly enhances the transcript level of pri-miR156e [67]. In addition, exogenous Os-miPEP156e does not affect the expression of other miRNAs. Interestingly, when exposed to Cd stress, the application of synthetic Os-miPEP156e reduces inhibition of seedling growth resulting in an increase in root length and biomass [67]. Moreover, Os-miPEP156e knockout mutants are more sensitive to Cd stress and have significantly lower plant height, root length and biomass than the wild-type.

Five additional synthetic rice miPEPs are found to increase the expression of their pri-miRNAs and improve the resistance of rice seedlings against Cd stress. These peptides include Os-miPEP172b (9 aa), Os-miPEP528 (25 aa), Os-miPEP396c (39 aa), miPEP171c (6 aa) and miPEP166b (31 aa) [67] (Table 2). 

Two more plants of the family *Poaceae* have been shown to encode miPEPs. First, pri-miRNAs444 of *Hordeum vulgare* is associated with ribosomes in barley shoots and roots, and translation of these miRNA precursors results in the synthesis of Hv-miPEP444a (119 aa) and Hv-miPEP444c (168 aa). CRISPR-cas9-mediated gene editing of Hv-miPEP444c coding sequence has revealed a reduction of root (30%) and shoot (20%) surface area in mutated plants compared to the wild type. CRISPR-cas9 mutagenesis of Hv-miPEP444a gives rise to lethal mutations [68]. Second, bioinformatic tools have predicted two miPEPs in *Zea mays* plants. These experimentally undescribed peptides have been revealed by mass spectrometry and include Zma-miPEP159d, in which miRNA is involved in the degradation of chlorophyll, and Zma-miPEP2275d, in which miRNA plays an important role in anther development and affects drought tolerance [69] (Table 2).

## 3. Subcellular and Molecular Targets of Plant miPEPs

To explore whether synthetic plant miPEPs after the external application and entering cells can target specific subcellular compartments, these peptides are labeled with fluorescent compounds. For example, grapevine Vvi-miPEP171d1 has been labeled by fluorescein isothiocyanate [63]. It has been shown that grape embryogenic calli treated with FITC-miPEP171d1 contain the fluorescence positioned mainly in the cytoplasm and partially in the nucleus, while the fluorescence signal in the cells treated with FITC is evenly distributed throughout the cells vvi-miPEP171d1 [63]. However, studies of kinetics of At-miPEP165a uptake into Arabidopsis roots have shown that the *in planta* mobility of synthetic At-miPEP165a fused to fluorescein (miPEP165a-FAM) results in even distribution inside the root cell [59]. 

Our studies of *B. oleraceae* Bo-miPEP156a have predicted that the peptide is mostly alpha-helical and possesses a nuclear localization signal (NLS) [61,66]. Indeed, this synthetic peptide is mainly found in plant and animal cells following exogenous application [61]. Significant accumulation in the nucleus has also been revealed for the *M. truncatula* Mt-miPEP171b after heterologous expression in tobacco leaf cells [70]. 

A poorly studied aspect of the miPEP activity is the identification of molecular targets that bind these peptides in plant cells. Generally, the targeting molecules could represent proteins, as it has been shown for lncRNA-encoded peptide ENOD40 (see Section 2.1; [35]), or nucleic acids. Indeed, we have predicted that cabbage Bo-miPEP156a could bind nucleic acids and experimentally revealed *in vitro* binding of this peptide to plant chromosomal DNA as well as chromatin [61]. *In planta* interaction of peptides Mt-miPEP171b and sequence-unrelated Mt-miPEP171b2 to their common nascent pri-miRNA molecules has been shown by microscopic method (FRET-FLIM) and isothermal titration calorimetry. It has been suggested that this kind of protein–RNA binding strongly depends on the presence of a specific linear set of codons encoding interacting miPEP, but not a strongly conserved nucleotide sequence, so that non-conserved miPEPs can perform specific regulatory functions on their pri-miRNA species only [30,55].

## 4. Possible Molecular Mechanisms Supporting Activity of Plant miPEPs

Assuming the peculiarities of interaction between miPEPs and their nascent pri-miRNAs [30,55], the following scheme for activation of pri-miRNAs by the encoded peptides has been proposed: miPEP translates in the cytoplasm from the full-length pri-miRNA or its fragment, containing ORF encoding miPEP; then, this peptide migrates to the nucleus, where it binds to the nascent pri-miRNA in the area of its coding sequence; such interaction further activates synthesis of the miRNA transcript at the transcription level [30,55]. However, the authors stress that there are some unanswered questions concerning the proposed mechanism. Particularly, “Through which mechanisms is miPEP-induced pri-miRNA expression regulated? Is the only interaction between miPEPs and their corresponding miORFs sufficient? Are regulatory proteins necessary for miPEP action and/or specificity, and, if so, which ones?” [30]. 

Evidently, it cannot be excluded that miPEPs have the potential to interact not only with RNA but also with DNA in the miRNA gene promoters or the Mediator protein complex. Thereby, miPEPs may regulate RNA polymerase II activity and/or that of the mediator complex in the initiation of transcription [29,30]. Intriguingly, these suggestions indicate that many different miPEPs (differing in length and sequence) retain the ability to form such complexes. Nevertheless, it is indirectly supported by the fact that the mediator complex can interact with thousands of species of transcription factors present in plants [30]. 

It is still unclear what kind of molecular targets are required by miPEPs to interact with their coding sequences in nascent chains of pri-miRNA. Currently, the existing experimental background does not allow us to conclude whether miPEP physically binds naked nascent pri-miRNA (or RNA in RNP form) or RNA-DNA hybrids [30,55]. Importantly, RNA-DNA hybrids in three-stranded R-loops involving displaced non-template ssDNA chains, template ssDNA chains and complementary nascent RNAs, and occupying 100–2000 base pairs can be frequently formed near the transcription start site regions of miRNA genes. Generally, it is important to note that R-loop accumulation can affect the transcription of nearby genes, altering the chromatin landscape and causing global changes in gene expression [71,72]. 

In some cases, the R-loops, positioned close to the transcription start site region of miRNA genes, may repress the initiation of transcription by blocking transcription factor binding at promoters [71,72]. We have proposed that miPEP binding to the own ORF in the nascent 5′-terminal pri-miRNA region after recognition of RNA or RNA/DNA hybrid may have an effect of helicase to melt of RNA-DNA hybrids in R-loop and, thus, increase transcription initiation and/or elongation [29]. 

Alternatively, R-loop destabilization may result in chromatin compaction around the promoter; thus, in a reduction in the binding of transcriptional activators [73]. This result suggests a scenario when miPEP binding to RNA/DNA hybrid may enhance transcription initiation by stabilizing R-loops. To verify these alternative hypotheses, experimental efforts are required to reveal true nucleic acid targets for plant miPEPs.

## 5. Chemically Synthesized Plant miPEPs and Potential Improvement of Agronomic Traits

It has been commonly accepted that due to successful peptide application through watering or spraying of plants, synthetic miPEPs are perspective alternatives to the use of chemicals in agronomy at field trials and in greenhouses. MiPEPs can have different agronomical significance and mediate important plant developmental aspects including enhanced root growth, early flowering, increased stem height, biotic and abiotic stress resistance, and, as a result, yield enhancement (Figure 1). Indeed, root modifications have been found after external application of synthetic At-miPEP165a, Bo-miPEP156a and Mt-miPEP171b in genera *Arabidopsis*, *Brassica* and *Medicago*, respectively (see Section 2.2.1 and Section 2.2.3). Similarly, Vvi-miPEP171d1 also regulates adventitious root formation in grapevine; thus, it may provide a very useful instrument in overcoming the bottlenecks of clonal propagation of this economically important crop (Section 2.2.4). Chemically synthesized At-miPEP166g and At-miPEP397a can increase the total root mass (Section 2.2.3), whereas Gm-miPEP172c is known to promote nodulation with the increase of nodule number (Section 2.2.1). Additionally, it has been found that synthetic Bo-miPEP397a and Bv-miPEP164b) have been revealed to increase total plant size and the foliar surface (Section 2.2.3). 

Potentially agronomical important miPEPs (At-miPEP858a and At-miPEP408), which are regulated by light *via* shoot-to-root mobile transcription factor HY5, have been recently reported in *A. thaliana* [74,75]. Extensive studies on the impact of these miPEPs have not been carried out to date, although it is known that the application of synthetic At-miPEP858a to growing *A. thaliana* plants results in early bolting and a significant increase in the plant height [41].

Considering physiological modifications under abiotic stress, it should be noted that (i) synthetic At-miPEP408 increases the sensitivity of seedlings toward low sulfur and arsenite As(III) stresses after addition to the growth media (Section 2.2.3). Conversely, synthetic Vvi-miPEP172b and Vvi-miPEP3635b, when applied to the grape plantlets, increase cold tolerance (Section 2.2.4). Likewise, the application of synthetic Os-miPEP156e reduces the inhibition of rice seedling growth after Cd stress. The synthetic rice peptides Os-miPEP172b (9 aa), Os-miPEP528 (25 aa), Os-miPEP396c (39 aa), miPEP171c (6 aa) and miPEP166b (31 aa) also increase resistance to Cd stress (Section 2.2.7) (Figure 1).

Increasing resistance to biotic stress has also been described for several plant miPEPs (Figure 1). It has been demonstrated that the exogenous application of *Phaseolus vulgaris* miPEPs Pv-miPEP169h, Pv-miPEP169k and Pv-miPEP169p to bean leaves results in increased resistance to necrotrophic fungus [57] (Section 2.2.1). Similarly, *A. thaliana* miPEPs At-miPEP169c, At-miPEP169h and At-miPEP396b can stimulate significant resistance to the necrotrophic fungus *Botrytis cinerea* [57]. Exogenic application of tomato Sl-miPEP169d also gives rise to the protective effect on the tomato against bacteria *Pseudomonas syringae* and *Xanthomonas* sp., as well as fungus *Alternaria solani* [57]. Four strawberry peptides (Fv-miPEP169h, Fv-miPEP169l, Fv-miPEP396a and Fv-miPEP396f) significantly increase plant fruit resistance after leaf inoculation with *B. cinerea* spores [57]. 

## 6. Phenomenon of Complementary Peptides (cPEPs) in Plants and Its Potential Use in Biotechnology

Strikingly, recent studies have revealed that chemically synthesized peptides having sizes from 5 to 40 residues and corresponding to the N-terminal parts of conventional plant proteins or luciferase are capable of moderately increasing the translation of their mRNAs in peptide-treated leaves and seedlings [76]. It seems that these peptides, called cPEPs, recognize their own coding regions in the 5′-terminal areas of mRNAs and somehow interact with ribosomes to increase ribosome recruitment at translation initiation sites. Most importantly, irrespective of the fine molecular mechanism of cPEP phenomenon, cPEPs could be used for biotechnology to improve crop quality and yields. Indeed, treatment of plants with specific cPEPs can improve plant resistance to *B. cinerea*, increase tolerance to heat stress and generally improve plant growth [76,77]. Moreover, a mixture of the selected specific cPEPs decreases the growth of some invasive and problematic weeds [76]. 

## 7. Conclusions

The above-mentioned data on the coding potential of plant pri-miRNAs still leave some important questions unanswered. Particularly, transcription activation of miRNA genes by plant miPEPs relates to the novel phenomenon in the area of nucleic acid-protein interactions based on the interaction of the peptides with their own coding ORFs. This previously unreported mechanism has been described as a specific binding of the peptide molecule to the RNA region, having only a specific linear set of cognate codons encoding this peptide. The main points to be clarified in connection to this phenomenon are the following: (i) which mechanism (initiation or elongation) is modified by miPEPs interacting with nascent chains of transcribing pri-miRNAs to increase the efficiency of transcription; (ii) are the molecular sensors (for example, proteins) recognizing miPEP–RNA interaction events to increase the efficiency of transcription? 

Since the cPEP phenomenon also relates to peptide recognition of the coding regions and subsequently increasing the translation initiation (see above), it is quite interesting if this mechanism is also involved in the enhancement of miPEP translation from the template pri-miRNA. Evidently, the above fundamental questions require intensive molecular studies in the future. Moreover, such studies may considerably contribute to possible future applications of artificial miPEPs and cPEPs in agriculture and biotechnology.

## Figures and Tables

**Figure 1 plants-13-01137-f001:**
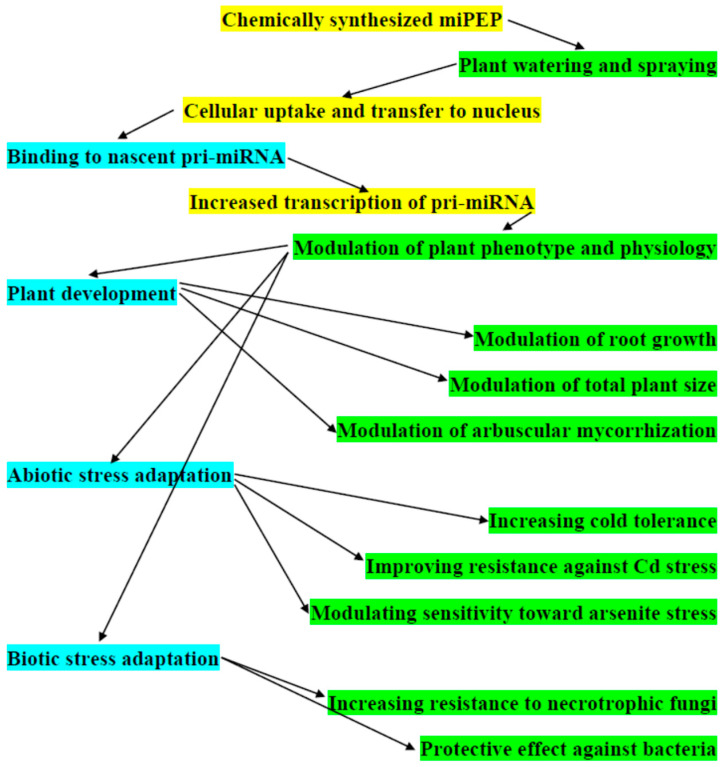
Generalized scheme of molecular and physiological activities related to the potential agronomic applications of plant miPEPs.

**Table 1 plants-13-01137-t001:** List of most miPEPs found in dicotyledonous plants.

Plant Family, Plant Species, miPEP Name and Size	Molecular and Sub-Cellular Targets of Synthetic Plant miPEPs	Biological Activity of Synthetic Plant miPEPs	Reference
***Fabaceae*** *Medicago truncatula* Mt-miPEP171b **20 aa** *	GFP-fusion localizes in the nucleus and cytoplasm after expression in tobacco leaf cells	activates transcription of pri-miR171b, increases expression of plant gene LOM1, stimulates arbuscular mycorrhization, reduces lateral root density	[53]
*Medicago truncatula* Mt-miPEP171a **10 aa**	unknown	decreases mycorrhization rate and LOM1 expression	[54]
*Lotus japonicus* Lj-miPEP171b **22 aa**	unknown	increases expression of pri-miRNA and mycorrhization rate	[54]
*Glycine max* Gm-miPEP172c **16 aa**	unknown	stimulates pri-miR172c expression, significantly increases the nodule number observed in plants	[56]
*Phaseolus vulgaris* Pv-miPEP169h **23 aa**	unknown	significant resistance to necrotrophic fungus *Botrytis cinerea*	[57]
***Sapindaceae*** *Dimocarpus longan* Dl-miPEP166 S338 **50 aa**	unknown	increases the expression of own pri-miRNA and inhibits plant gene *ATHB15*	[58]
***Brassicaceae*** *Arabidopsis thaliana* At-miPEP165a **18 aa**	mostly cytoplasm	increases expression of pri-miRNA and increases primary root length due to the stimulation of cellular proliferation rather than an increase in cell length	[53,59]
*A. thaliana* At-miPEP397a **18 aa**	unknown	increases the expression of their cognate pri-miRNAs and total root development	[60]
*A. thaliana* At-miPEP858a **44 aa**	Unknown	enhances the transcription of the corresponding pri-miRNA and causes increased root length and modulated levels of flavonoids	[41]
*Brassica oleraceae* Bo-miPEP156a **33 aa**	mostly nucleus in plant and animal cells	enhances the transcription of the corresponding pri-miRNA and causes increased root length	[55,61]
*B. oleraceae* Bo-miPEP397a **10 aa**	unknown	increases the expression of their respective pri-miRNAs and increases total plant size	[60]
*A. thaliana* At-miPEP408 **35 aa**	unknown	increases the sensitivity of seedlings toward low sulfur and arsenite As(III) stresses	[62]
***Vitaceae****Vitis vinifera* Vvi-miPEP171d1 **7 aa**	mainly in the cytoplasm and partially in the nucleus	increases expression of Vvi-miR171d and pri-miR171d and the number of adventitious roots	[63]
*Vitis vinifera* Vvi-miPEP164c **16 aa**	unknown	the upregulated transcript levels of pre-miR164c and inhibited proanthocyanidin synthesis	[64]
*V. vinifera* Vvi-MIR172b **16 aa**	unknown	increases the expression of corresponding miRNAs; plant tissue exhibits a higher cold tolerance	[65]
***Solanaceae****Solanum lycopersicum* Sl-miPEP169d **10 aa**	unknown	exogenic application lowers the level of tomato leaf infection by *Alternaria solani*	[57]
***Rosaceae****Fragaria vesca* Fv-miPEP169h **21 aa**	unknown	significantly decreases lesion size after leaf inoculation with *B. cinerea* spores	[57]

* aa—means amino acids. Plant family names and the size of peptides are given in bold.

**Table 2 plants-13-01137-t002:** List of most miPEPs found in monocotyledonous plants.

Plant Family, Plant Species, miPEP Name and Size	Molecular and Sub-Cellular Targets of Synthetic Plant miPEPs	Biological Activity of Synthetic Plant miPEPs	Reference
***Poaceae****Oryza sativa* Os-miPEP156e **36 aa**	unknown	enhances the transcript level of pri-miR156e and reduces inhibition of seedling growth resulting in an increase in root length and biomass under Cd stress	[67]
*Oryza sativa* Os-miPEP172b **9 aa**	unknown	increases the expression of pri-miRNA and improves the resistance of rice seedlings against Cd stress	[67]
*Hordeum vulgare* Hv-miPEP444c **119 aa**	unknown	CRISPR-cas9 mediated gene editing of Hv-miPEP444c coding sequence has revealed that a reduction of root and shoot surface area	[68]
*Zea mays* Zma-miPEP159d	unknown	unknown	[69]

Plant family names and the size of peptides are given in bold.

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
