# Peer review of "Biological Activity of Artificial Plant Peptides Corresponding to the Translational Products of Small ORFs in Primary miRNAs and Other Long “Non-Coding” RNAs"

_plants, 2024, doi:10.3390/plants13081137_

Round 1

Reviewer 1 Report

Comments and Suggestions for Authors

In this manuscript, the authors have discussed about the discovery of peptides (lncPEPs)

produced by long non-coding RNAs (lncRNAs) in various plant species. It highlights the

functions of various small peptides encoded by Pri-miRNAs in diverse plant families, mainly in

flowering plants. Authors have also discussed about functions of chemically synthesized small

peptides when applied exogenously on different plant species. Specifically, authors have focused

on the regulatory functions of synthetic miPEPs and the necessity for extensive studies to explore

their activities and corresponding ORFs. Authors have also discussed about the functions of

complementary peptides.

The scientific content of the present study is useful for understanding how novel small peptides

encoded by non coding RNAs regulate essential traits of the plants. This study will also help to

identify the potential small peptides that can be genetically engineered to enhance important

agronomic traits in plants. However, the manuscript needs extensive revision for typographical

errors and grammar. The manuscript needs to address the following major concerns:

Text

Authors are suggested to change the title as in text they have majorly discussed about peptides

encoded by miRNAs but they have mentioned in title only about long-non coding RNAs.

Revise the whole abstract as many lines require reframing for instance,

“Evidently……..organisms”. Also, avoid using larger sentences like, “Studies of

chemically………….protein products.

Authors are suggested to avoid the use of long keywords, “chemically synthesized micropeptide,

transcription of pri-miRNA, microRNA primary transcripts, miPEP and biotechnology”. Replace

them with short meaningful words.

Authors are advised to once recheck the whole text for grammar mistakes.

Authors are suggested to form different subsections for lncRNA and miRNA encoded peptides.

Author should add conclusion of the article.

Authors should discuss about the future perspectives in detail for research related to small

peptides as this is the emerging area of research.

Figures and tables

In table 1 and 2, authors are suggested to make different column for number of amino acids of

small peptides of different families.

Authors are suggested to add a figure depicting various functions of small peptides in different 

plant species.

Comments on the Quality of English Language

NA

Author Response

1) Authors are suggested to change the title as in text they have majorly discussed about peptides encoded by miRNAs but they have mentioned in title only about long-non coding RNAs.

It has been done (see Title, marked in yellow)

2) Revise the whole abstract as many lines require reframing for instance,

“Evidently……..organisms”. Also, avoid using larger sentences like, “Studies of

chemically………….protein products.

It has been done (see Abstract, lines 17-18, 20-21, marked in yellow)

3) Authors are suggested to avoid the use of long keywords, “chemically synthesized micropeptide, transcription of pri-miRNA, microRNA primary transcripts, miPEP and biotechnology”. Replace them with short meaningful words.

It has been done (see Keywords, lines 23-24, marked in yellow)

4) ...the manuscript needs extensive revision for typographical errors and grammar. Authors are advised to once recheck the whole text for grammar mistakes.

It has been done (see lines 52-53, 113, 159, 168, 183, 189-190, 214, 221-222, 247-248, 255-256, marked in yellow)

5) Authors are suggested to form different subsections for lncRNA and miRNA encoded peptides.

These items are in different subsections (see lines 66 and 138).

6) Author should add conclusion of the article. Authors should discuss about the future perspectives in detail for research related to small peptides as this is the emerging area of research.

Conclusions have been added to discuss about future perspectives (see lines 462-479, marked in yellow).

7) In table 1 and 2, authors are suggested to make different column for number of amino acids of small peptides of different families.

There is no need in different column since the peptide sized have been marked in bold (see Tables 1 and 2, marked in yellow).

8) Authors are suggested to add a figure depicting various functions of small peptides in different plant species.

All required information is presented in Tables 1 and 2.

Reviewer 2 Report

Comments and Suggestions for Authors

This is a well-written manuscript describing the role of short peptides stemming from lncRNAs in various physiological and pathological responses of various plants.

My main suggestion is to prepare a figure on known and potential mechanisms of function of these peptides.

There are some minor things, mostly in the Abstract to be addressed - I attached a pdf with changes.

Author Response

1) My main suggestion is to prepare a figure on known and potential mechanisms of function of these peptides.

All required information is presented in Tables 1 and 2. Moreover, our attempts to prepare figures in early versions of the manuscript gave less informative picture because of too many peptides to compare.

2) There are some minor things, mostly in the Abstract to be addressed - I attached a pdf with changes.

It has been done (see lines 12, 14, 21, 36, 141-143, marked in yellow).

Reviewer 3 Report

Comments and Suggestions for Authors

The document is a review about untranslated miRNAs and their precursors, as well as their effect on expression in plants.

It is a topic of interest to the community and is presented in a very descriptive way, although it provides new information.

It would be advisable to include some diagram with the possible mechanism of action of some of these minimolecules.

Author Response

1) It would be advisable to include some diagram with the possible mechanism of action of some of these minimolecules.

All required information is presented in Tables 1 and 2.

Round 2

Reviewer 2 Report

Comments and Suggestions for Authors

My comments are addressed.

Author Response

My comments are addressed.

- Thanks to referee. No actions are needed.

Reviewer 3 Report

Comments and Suggestions for Authors

After review the paper, I still yhionking some graphical evidence must be shown

Author Response

After review the paper, I still yhionking some graphical evidence must be shown.

- Graphical part has been included into section 5. See also lines 419, 449 and 451.